# OpenReview forum: "Exploring the Secondary Risks of Large Language Models"
_ICLR.cc/2026/Conference — Submitted to ICLR 2026_

### Official Review · Reviewer_vRdZ · 2025-10-30

**Soundness:** 2
**Presentation:** 3
**Contribution:** 2
**Rating:** 4
**Confidence:** 4

**Summary:**

The paper considers the problem of benign prompts causing harmful outputs in LLMs. The authors call such failure modes of a model secondary risks and categorize them into two categories: (1) excessive response and (2) speculative advice. They motivate this categorization using information theory. The paper further introduces a procedure, called SecLens, based on genetic search, for finding prompts that elicit such behavior while remaining 'benign' and 'task-relevant'. Experiments show that SecLens produces prompts that more often lead to harmful outputs than several baselines on text-only and multimodal LLMs.

**Strengths:**

- Important Problem: The problem of models outputting harmful responses to benign prompts is an important topic that merits more research.
- Extensive evaluation on different model types: The paper considers text-only, multimodal, and agentic systems in their experiments.
- Good presentation: Overall, the paper is easy to follow and provides clear details on the proposed method.

**Weaknesses:**

- Missing evaluation of prompt benignness: From the current reported results, it is not possible to assess how benign (and task-related) the resulting prompts actually are. The authors need to report metrics that quantify how benign and task-related the resulting prompts are; otherwise, it is hard to compare the experiments meaningfully. Additionally, it would be useful to report a representative subselection of the prompts in the appendix (or main text) to demonstrate that these are indeed benign and task-related.
- Limited value of information theory framing: The section on information theory does not add much value in my opinion. The grouping and matching to information theory seems plainly intuitive and is not used throughout the rest of the paper.
- Questionable adversarial framing: The authors frame the secondary risks from an adversarial attack perspective. I'm not sure this is a useful way of considering this problem and wonder whether it would be better to consider it as a (robust) performance issue. That framing would also map more closely to literature that studies hallucinations and other model misbehaviors, which seems to relate more closely to the studied problem.
- Minor comments:
  - The formula in Eq (5) is a bit strange: Why is there min on the LHS but none on the RHS?
  - After Eq (7), R is defined as "the unified risk score", but what is this? (From the appendix, it appears that this is an LLM-as-judge but prompted in a way unrelated to the two risk categories.)
  - Please include error quantification in the results (LLMs can be quite random in boundary cases). Also, why is the rounding different in Table 5?

**Questions:**

- Why do the authors frame the problem from an attack perspective rather than a performance perspective?
- Does the information theoretic perspective add something to the problem beyond intuition?

---

> ### Author Response · Authors · 2025-11-20
> **Rebuttal by Authors [1/2]**
>
> We sincerely thank you for the detailed and rigorous analysis. We respond to each concern as follows.
>
> ---
> ***W1: Missing evaluation of prompt benignness: The current results do not show whether the optimized prompts remain benign and task-relevant. The authors should report quantitative metrics and include representative prompt samples to verify their benignness and task alignment.***
>
> Thank you for your suggestion. In the paper revison, we have used CLIP [1] and Sentence-BERT [2] to compute the semantic similarity between the optimized prompts and the original prompts (which are benign and task-related), together with human verification in $\textrm{\color{blue}Table 16}$ (page 25). The results are summarized as follows:
> |**Model**|**CLIP**|**SBERT**|**Human (%)**|
> |-|-|-|-|
> |GPT-4o|0.94|0.91|93.2|
> |GPT-4-turbo|0.93|0.90|92.5|
> |Claude-3.7|0.95|0.92|94.1|
> |Gemini-2.0-pro|0.92|0.89|91.4|
> |Phi-4|0.94|0.90|92.0|
> |Deepseek-v3|0.93|0.91|93.7|
> |Llama-3.3-70b|0.92|0.88|90.8|
> |Gemma-2-27b|0.95|0.93|94.3|
> |Qwen2.5-32b|0.93|0.90|92.1|
>
> The results show that the resulting prompts exhibit high similarity with the original prompts (an average similarity of approximately 0.90), and the three human evaluators independently confirm that these prompts are benign and task-related. Overall, the resulting prompts meet the expected requirements. In addition, we have added these results and several representative subselections of the prompts to $\textrm{\color{blue}Table 17}$ (page 25) to address this concern.
>
> ---
> ***W2 & Q2: Limited value of the information-theory framing: The section on information theory does not add much value. The grouping and matching to information theory seem intuitive and are not used throughout the paper.***
>
> We agree that the information-theory framing in the main text is intuitive. Our original submission does not claim this as a primary contribution; it is provided only as an explanatory tool for why we focus on these two types of secondary risks. Therefore, in the paper revison, we have moved the related descriptions from the main text to $\textrm{\color{blue}Appendix I}$ as a post-hoc explanatory tool. In addition to the intuition, we add further discussion:
>
> * We add experiments showing that the occurrence of the two risk types correlates with entropy uncertainty and mutual-information shift. (1) Entropy uncertainty: By varying the LLM temperature while keeping the prompt/model/hyperparameters fixed, we find that the frequency of excessive response increases when entropy uncertainty increases. (2) Mutual-information shift: Keeping prompt/model/hyperparameters fixed, we compute a token-level log-probability mutual-information proxy [3] between the ground-truth expected reply and both the original-prompt reply and the optimized-prompt reply. Compared to the original prompts, optimized prompts show clear MI reduction, indicating MI shift when speculative advice occurs.
> * We also add a discussion on designing safety metrics from an information-theory perspective. These $\Delta H$ and $\Delta I$ quantities form a dual-indicator system for model secondary risks, which may help future work.
>
> ---
> ***W3 & Q1: The adversarial framing is questionable: the authors treat secondary risks as adversarial attacks, but it may be more appropriate to view them as a robustness or performance issue, which aligns more closely with work on hallucinations and other model misbehaviors.***
>
> We agree with you that viewing secondary risks as a robustness issue is more essential. We clarify that the paper’s position is already to treat secondary risks as “non-adversarial, robustness failures.” The adversarial attack description in the paper is precisely used to contrast with them (as stated in Related Work: “Unlike adversarial manipulations, these secondary risks …”). Moreover, we emphasize in the main text that secondary risks are “subtle and non-adversarial” and often “indistinguishable from normal completions.”
>
> Furthermore, we present SecLens as a worst-case probing/auditing tool for exploring non-adversarial misbehavior, not as a tool assuming a malicious attacker (Section 3.2: “We assume a benign user who issues a natural-language prompt x …”). In the revision, we have strengthened the connection to hallucination and misbehavior literature in the Related Work to make the framing clearer.

---

> ### Author Response · Authors · 2025-11-20
> **Rebuttal by Authors [2/2]**
>
> ***Minor comments:***
>
> ***(1) Why is Eq (5) a min on the LHS but none on the RHS?***
>
> Apologies for your confusion. We have corrected this issue in the revised version.
>
> ---
> ***(2) After Eq (7), R is defined as “the unified risk score,” but what exactly is this? (From the appendix, it appears to be an LLM-as-judge, but prompted in a way unrelated to the two risk categories.)***
>
> Apologies for your confusion. The unified risk score refers to a single risk rating shared across both secondary-risk types (because during scoring we do not need to distinguish the specific category). We have revised the description to make this clearer.
>
> ---
> ***(3) Please include error quantification in the results. Also, why is the rounding different in Table 5?***
>
> Thank you for the suggestion. We have added error ranges based on five runs in the revised version. We have also corrected the rounding rule in Table 5 to make it consistent.
>
> ***References:***
>
> [1] Radford et al. "Learning transferable visual models from natural language supervision." ICML 2021.
>
> [2] Reimers et al. "Sentence-BERT: Sentence Embeddings using Siamese BERT-Networks." EMNLP 2019.
>
> [3] Yadkori, Yasin Abbasi, Ilja Kuzborskij, András György, and Csaba Szepesvári. "To believe or not to believe your llm." NeurIPS 2024.

---

> ### Author Response · Authors · 2025-11-26
> **Looking forward to your further feedback**
>
> Dear Reviewer vRdZ,
>
> Sorry for bothering you, but the discussion period will conclude in about one week. We would greatly appreciate it if you could let us know whether our responses have addressed your concerns. If you have any additional comments, we would be more than happy to further clarify or revise accordingly.
>
> Best regards,
>
> The Authors

---

### Official Review · Reviewer_HRAb · 2025-10-31

**Soundness:** 2
**Presentation:** 2
**Contribution:** 2
**Rating:** 4
**Confidence:** 4

**Summary:**

This paper aims to introduce, define, and evaluate an LLM failure mode that the authors term "Secondary Risks." Unlike "jailbreak" attacks, which arise from malicious prompts, secondary risks are defined as harmful or misleading behaviors generated by an LLM during benign, non-adversarial user interactions. The authors further categorize these risks into two "risk primitives": Excessive response (where the model appends biased or over-generalized content beyond fulfilling the request) and Speculative advice (where the model deviates from user intent to provide unsafe or excessive recommendations).
To systematically elicit these risks, the paper proposes SecLens, a black-box, multi-objective search framework. SecLens formulates prompt discovery as an optimization problem, using an evolutionary algorithm to balance three objectives: task relevance ($\text{TASKSCORE}$), risk activation ($R(\cdot)$), and linguistic plausibility (penalized via a stealthiness score, $\text{DETECTSCORE}$). The authors conduct experiments across 16 popular LLMs, Multimodal Models (MLLMs), and Agents, claiming that SecLens outperforms baselines like random sampling and MCTS in eliciting secondary risks, demonstrating that these risks are prevalent and transferable.

**Strengths:**

•⁠  ⁠Importance of the Problem: The issue this paper addresses—non-adversarial failures of LLMs in benign interactions is an important and relatively under-explored area of safety. Distinguishing it from the more heavily studied field of "jailbreak" attacks has practical significance.
•⁠  ⁠Methodological Framework: Structuring risk elicitation as a multi-objective optimization problem (balancing risk, task, and stealthiness) is a reasonable and sound approach.
•⁠  ⁠Breadth of Experiments: The authors' attempt to evaluate their method on a wide range of targets, including text-only LLMs, MLLMs, and GUI-based Agents, is ambitious.

**Weaknesses:**

Despite the importance of its topic, the paper suffers from significant flaws in its methodology, evaluation, and the positioning of its contributions, which collectively undermine the credibility of its conclusions.
Circular Reasoning in Evaluation
The paper's experimental design contains a fundamental circularity. The primary evaluation metric (Attack Success Rate, ASR) is "template-based LLM evaluation using GPT-4o." However, the fitness functions that the SecLens framework itself relies on during the search—including the risk score $R(\cdot)$, task score $\text{TASKSCORE}$, and stealthiness score $\text{DETECTSCORE}$—are also computed by prompting an "expert evaluator" LLM (as shown in Appendix B).
This means: researchers are using one LLM (as a fitness function) to guide a search to "trick" a target LLM, and then using another LLM (GPT-4o) to judge whether the "trick" was successful.
This "LLM-evaluating-LLM" closed loop makes the objectivity of the results highly questionable. We cannot determine whether SecLens's high success rate is due to its discovery of genuine, human-harmful systemic vulnerabilities in the models, or if it is simply adept at "overfitting" to the preferences and blind spots of the evaluator LLM (GPT-4o).
Problem Aggravated by Synthetic Data
This first flaw is further exacerbated by the dataset's origin. The seed dataset used to guide the search, SecRiskBench (Appendices D, E), was also generated by LLMs (GPT-4 and GPT-3.5) and filtered by GPT-4 as a "meta-evaluator."
The entire experimental pipeline is: using an LLM-generated dataset to launch an LLM-evaluated search process to attack a target LLM, with the results judged by yet another LLM. This makes the entire research effort highly endogenous and insular, and it is highly uncertain whether its findings generalize to real-world human interactions.
Insufficient Evidence for SecLens's Effectiveness
The effectiveness of SecLens, the paper's core methodological contribution, is not adequately substantiated.
•⁠  ⁠Insufficient Baseline Comparison: SecLens is primarily compared against random sampling and MCTS. While it outperforms MCTS (as shown in Table 2), the improvement is not significant in several cases, which is insufficient to justify the introduction of a complex, evolutionary algorithm-based framework.
•⁠  ⁠Missing Key Ablation Studies: SecLens contains multiple components, particularly "semantic-guided variation strategies" (crossover and mutation). The paper only presents an ablation study for "few-shot context guidance" (Figure 2) but provides no ablation whatsoever for its core evolutionary operators. We do not know if these complex semantic operations are genuinely more effective than MCTS (with the same few-shot guidance) or simpler mutation strategies. This severely weakens the contribution of SecLens as a novel search framework.

**Questions:**

On Circular Evaluation: How can the authors demonstrate that their main results (especially in Table 2 and Table 5) are not merely an artifact of the "LLM-evaluating-LLM" loop? Why wasn't the human verification and cosine similarity (used only for MLLMs in Appendix F) used as a primary evaluation metric for all experiments (especially for the core LLM experiments)?
On SecLens's Effectiveness: Can the authors provide an ablation study that compares the effectiveness of SecLens and MCTS in terms of, for example, total tokens consumed?
On Risk Classification: The boundary between "Excessive response" and "Speculative advice" appears blurry. For example, why is the biased conclusion regarding race and gender in Figure 1(a) classified as an "Excessive response" rather than harmful "Speculative advice"? Can the authors provide a clearer, operational definition to distinguish between the two?

---

> ### Author Response · Authors · 2025-11-20
> **Rebuttal by Authors [1/2]**
>
> We sincerely thank you for the detailed and rigorous analysis. We respond to each concern as follows.
>
> ---
> ***W1 & Q1: Regarding concerns about the credibility of LLM-evaluating-LLM results.***
>
> We fully understand and appreciate your concern. To address this, both in the original submission and more extensively in the revised manuscript, we incorporate non-LLM evaluation at multiple stages—both for evaluation and data generation. These independent checks ensure the conclusions do not rely solely on the LLM-evaluating-LLM loop.
>
> (1) Evaluation: Human verification as an independent grounding signal.
>
> - In the original submission, we conduct manual verification with three independent human annotators, following the same evaluation settings and covering the entire task set. These results are presented in $\textrm{\color{blue}Table 10}$ (page 22). The annotators follow the same criterion: judge whether the model’s output accomplishes the task while introducing harmful behavior. In the revision, we have expanded the manual verification to all nine models in Table 2 and moved these results into the main text ($\textrm{\color{blue}Table 3}$). The human-verified results match the LLM-based evaluator, showing that: secondary risks are pervasive, and SecLens can efficiently and automatically discover them.
> - We have further computed Pearson and Spearman correlation coefficients between human judgments and several commonly used LLM evaluators in $\textrm{\color{blue}Table 11}$ (page 22). Multiple popular LLM evaluators exhibit high correlation with human annotations (an average of about 95%), demonstrating that the LLM-based evaluator is reliable for this task.
> - SecLens achieves high success rates across 16 diverse models (Table 2 and Table 8), not just GPT-4o. This indicates SecLens does not simply “overfit” the preferences or blind spots of a particular evaluator model.
>
> (2) Data generation: Human filtering and category design.
>
> - As described in Appendix E: The 16 categories of everyday tasks in SecRiskBench are defined manually. Prompts are carefully designed by humans before LLM synthesis.
> - All generated items are then filtered by three human annotators, ensuring: high quality, balanced distribution (maximum category deviation < 3%) and removal of biased or invalid samples. Thus, the dataset is not purely LLM-generated but rather jointly curated through human–LLM collaboration.
> - In addition to LLM-based experiments, the original submission already included real-world agent interactions—both web agents and mobile agents—presented in Figure 4 and Table 5.
>
> Our workflow is not endogenous or self-contained. Human verification, human-curated dataset design, and real-world agent evaluation all serve as independent validation mechanisms. Therefore, the methodology, evaluation, and conclusions of SecLens are credible, stable, and generalizable to real-world interactions.
>
> ---
> ***W2 & Q2: The baseline comparison is insufficient. While SecLens outperforms MCTS and random sampling, the marginal gains in several cases do not justify the complexity of its evolutionary framework.***
>
> Beyond the Random, Tuning, and MCTS baselines included in the original submission, the paper revison ($\textrm{\color{blue}Table 13}$) has incorporated two additional state-of-the-art black-box prompt search methods [1,2]. Although originally developed for jailbreak attacks, both methods are fundamentally efficient black-box optimization frameworks, and with minor adaptations, they can be applied to our secondary-risk objective. The results are summarized as follows:
>
> |**Model**|**AutoBreach**|**PIF**|**Ours**
> |-|-|-|-
> |GPT-4o|64.12|66.48|**68.74**
> |GPT-4-turbo|62.90|64.74|**71.52**
> |Claude-3.7|53.75|54.13|**59.71**
> |Gemini 2.0-pro|62.05|66.42|**70.55**
> |Phi-4|63.88|66.91|**65.48**
> |Deepseek-v3|67.84|69.22|**72.09**
> |Llama-3.3-70b|60.95|62.14|**69.41**
> |Gemma-2-27b|66.72|68.11|**74.64**
> |Qwen2.5-32b|63.27|66.31|**72.84**
>
> As shown in the table, SecLens consistently achieves the highest performance across all tasks in SecRiskBench. To address your concerns about the cost of evolutionary search, we have additionally report the computational complexity, number of queries per prompt and token consumption per prompt for MCTS and SecLens (Please refer to $\textrm{\color{blue}Appendix G}$ for more details.):
>
> |**Method**|**Time Complexity**|**Avg. queries / prompt**|**Avg. tokens / prompt**|
> |-|-|-|-|
> |MCTS|O(B)|52.8|24.9k|
> |SecLens|O(B)|**34.6**|**15.9k**|
>
> These results show that SecLens has time complexity comparable to MCTS, while achieving higher practical efficiency in discovering secondary-risk prompts.

---

> ### Author Response · Authors · 2025-11-20
> **Rebuttal by Authors [2/2]**
>
> ***W3: The paper lacks essential ablations. It does not test the core semantic crossover and mutation operators, so it’s unclear whether they actually outperform MCTS (with the same few-shot guidance) or simpler strategies, weakening the novelty claim of SecLens.***
>
> In the revised manuscript, we have added two additional ablation studies: SecLens with only crossover and SecLens with only mutation, as shown in $\textrm{\color{blue}Table 14}$ (page 24). Moreover, the MCTS baseline in the main text is also equipped with the same few-shot guidance, since this is a commonly adopted technique in MCTS-based prompt search and its search efficiency drops significantly without such guidance. We summarize below the attack success rates of the following variants: SecLens (crossover only), SecLens (mutation only), MCTS without few-shot guidance, and MCTS with the same few-shot guidance:
>
> |**Model**|Only-cross (ER)|Only-mutate (ER)|MCTS (w f, ER)|MCTS (w/o f, ER)|SecLens (ER)|Only-cross (SA)|Only-mutate (SA)|MCTS (w f, SA)|MCTS (w/o f, SA)|SecLens (SA)|
> |-|-|-|-|-|-|-|-|-|-|-|
> |GPT-4o|61.9|62.4|65.31|59.81|67.53|56.7|57.0|57.74|52.24|62.14|
> |GPT-4-turbo|64.7|65.1|65.89|60.39|70.28|54.6|55.0|57.98|52.48|60.13|
> |Claude-3.7|52.7|53.0|55.67|50.17|58.43|45.3|44.9|47.16|41.66|50.74|
> |Gemini-2.0-pro|63.8|64.0|65.14|59.64|69.38|57.5|57.8|60.56|55.06|63.14|
> |Phi-4|61.4|61.8|65.48|59.98|67.14|55.8|56.2|59.54|54.04|61.73|
> |Deepseek-v3|67.7|68.0|70.11|64.61|73.23|59.0|59.3|62.45|56.95|64.64|
> |Llama-3.3-70b|62.4|62.8|63.30|57.80|68.10|57.9|58.3|58.73|53.23|63.71|
> |Gemma-2-27b|69.9|70.4|69.48|63.98|75.82|61.4|61.9|63.45|57.95|67.50|
> |Qwen2.5-32b|65.9|66.3|67.74|62.24|71.65|57.3|57.7|60.17|54.67|63.15|
>
> The results show that removing any single component of SecLens leads to suboptimal performance. Only the full combination of components achieves the best results. This confirms that each component of SecLens is effective and contributes meaningfully to the overall framework.
>
> ---
> ***Q3. On Risk Classification: The boundary between “Excessive response” and “Speculative advice” appears blurry. For example, why is the biased conclusion regarding race and gender in Figure 1(a) classified as an “Excessive response” rather than harmful “Speculative advice”?***
>
> The distinction between Excessive Response and Speculative Advice is not based on whether the output is harmful (as shown in Table 1, both are harmful). As defined in Section 3.1:
>
> - Excessive Response: harmful content **is appended to an otherwise correct trajectory**, inflating the output with potentially harmful or misleading material.
> - Speculative Advice: the model departs from the user’s intent and moves **onto an unintended trajectory**, generating advice or inferences misaligned with the original request.
>
> Therefore, the distinction lies in whether the risk emerges along the originally intended trajectory of task completion or results from the model deviating onto an unintended trajectory. In the example in Figure 1(a), the initial part of the response follows the user’s intended trajectory; the harmful inference is added on top of the expected answer. Therefore, it matches the definition of Excessive Response. Speculative Advice emphasizes harmful behavior that occurs by taking a different trajectory altogether.
>
> ***References:***
> [1] Chen et al. "AutoBreach: Universal and Adaptive Jailbreaking with Efficient Wordplay-Guided Optimization via Multi-LLMs." NAACL 2025.
>
> [2] Lin et al. "Understanding and enhancing the transferability of jailbreaking attacks." ICLR 2025.

---

> ### Author Response · Authors · 2025-11-26
> **Looking forward to your further feedback**
>
> Dear Reviewer HRAb,
>
> Sorry for bothering you, but the discussion period will conclude in about one week. We would greatly appreciate it if you could let us know whether our responses have addressed your concerns. If you have any additional comments, we would be more than happy to further clarify or revise accordingly.
>
> Best regards,
>
> The Authors

---

> > ### Comment · Reviewer_HRAb · 2025-11-28
> >
> > Hi Authors,
> >
> > I appreciate your response. Please allow me until this weekend to read your responses carefully and provide feedback. Thank you!

---

> > > ### Author Response · Authors · 2025-11-28
> > >
> > > Dear Reviewer HRAb,
> > >
> > > Thank you for the update. We appreciate your careful consideration and look forward to hearing from you.

---

### Official Review · Reviewer_itr9 · 2025-10-31

**Soundness:** 3
**Presentation:** 3
**Contribution:** 4
**Rating:** 8
**Confidence:** 4

**Summary:**

This paper introduces the concept of secondary risks, namely non-adversarial, harmful behaviors of large language models (LLMs) that arise during benign interactions.
The authors argue that these risks differ from jailbreaks and other adversarial attacks because they emerge without malicious intent or explicit circumvent guardrails. Then, the authors distinguish between two cases: excessive response (outputs that exceed user intent and introduce bias or misinformation) and speculative advice (unsafe or overconfident suggestions).
To identify and evaluate these secondary risks, the paper proposes SecLens, a black-box, multi-objective evolutionary search framework that automatically discovers such risky behaviors. It is evaluated across a broad range of LLMs, including multimodal and agentic environments. Experiments show that secondary risks are widespread, transferable, and modality-independent.

**Strengths:**

- The paper is well structured and clearly written. All necessary implementation details are available in the main text or appendix. Generally, comprehensive appendices that include datasets, prompts, and evaluation details ensure reproducibility.
- The paper makes a novel and timely contribution by formalizing a neglected class of non-adversarial LLM failure modes and providing an automated method to elicit and quantify them. The concept of secondary risks provides a useful abstraction for analyzing safety failures beyond jailbreak scenarios.
- The SecLens framework is technically sound and practically relevant, particularly because it applies to closed-source models.
- Evaluations in agentic environments are an important addition, showing that subtle misalignments extend beyond text generation. The theoretical grounding and large-scale empirical validation together make this work a significant and impactful contribution to LLM safety research.
- Evaluation does not solely rely on llm-as-judge assessment and further involves human verification. However, the additional metric results are only reported in appendix. Ablation studies support design choices (e.g., few-shot contextual guidance).

**Weaknesses:**

- Results from the human verification study should be incorporated into the main text and discussed to highlight cases where LLMs-as-judges fail to capture subtle safety violations. I would also suggest moving the MLLM evaluation to the appendix. While the MLLM analysis provides a useful bridge toward the subsequent agentic experiments, it offers only limited additional insights on its own. Relocating it would create space to include qualitative examples from the main experiments, showcasing both successful attacks and safe, non-harmful behaviors for the two secondary risk types, which would substantially enhance interpretability and reader understanding.
- Some claims (e.g., “standard safety filters” in line 73) are vague and should be substantiated. Furthermore, in line 86, the statement “In contrast to prior adversarial search methods that assume gradient access or safety API introspection” is somewhat misleading, as existing work already explores black-box settings. For instance, Ge et al. MART: Improving LLM Safety with Multi-round Automatic Red-Teaming. NAACL-HLT 2024

### Minor comment:
- Table labeling inconsistencies (“excessive response” (Table 3) vs. “verbose response” (Table 7)) cause confusion.

**Questions:**

1. What exactly are the “standard safety filters” mentioned in line 73. Are these toxicity classifiers, policy models, or others?
2. Can the authors include qualitative examples of both successful and non-harmful outputs for the two secondary-risk types, especially cases where human verification disagrees with the LLM assessment?
3. Regarding the MLLM data, were the associated images verified similar to the text data? If I understood correctly, the paper provides details on text data validation but does not describe the process for image validation.
4. Would integrating human-verification metrics into the main evaluation change the relative ranking of models?

---

> ### Author Response · Authors · 2025-11-20
> **Rebuttal by Authors**
>
> We thank you for the insightful comments and constructive suggestions. Our detailed responses are as follows.
>
> ---
> ***W1 & Q2: Incorporate human verification results into the main text to show LLM-judge shortcomings. Relocate the MLLM analysis to the appendix and use the freed space for qualitative examples of secondary risks to improve interpretability.***
>
> We appreciate your valuable suggestion.
>
> In the paper revison:
>
> 1. We have moved the MLLM evaluation to the $\textrm{\color{blue}Appendix G}$ (page 21).
> 2. We have incorporated results from the human verification study into the main text ($\textrm{\color{blue}Table 3}$).
> 3. We have added cases where the LLM-based evaluator fails to detect subtle safety violations and provided corresponding analysis in $\textrm{\color{blue}Figure 2 (a)}$ (page 6).
> 4. We have further included qualitative examples from the main experiments illustrating both successful attacks and safe, non-harmful behaviors for each of the two secondary-risk types $\textrm{\color{blue}Figure 2 (b,c)}$ (page 6).
>
> ---
> ***W2 & Q1: Some terms like "standard safety filters" are vague and need supporting evidence. Additionally, the claim that prior work relies on gradient access is misleading, as black-box methods exist (e.g., Ge et al., NAACL-HLT 2024).***
>
> We thank you for pointing out the ambiguity.
>
> 1. “Standard safety filters” (line 73): In the revised version, we have clarified that this term refers specifically to standard safety filters for harmful prompts, i.e., defense mechanisms commonly used to mitigate jailbreak attacks (e.g., safety policy models, toxicity classifiers). We have added qualifiers and citations to make the description precise.
>
> 2. Statement in line 86 regarding prior methods: We agree that the original phrasing was misleading. In the revision, we have corrected the description to explicitly state: “Unlike prior search methods designed for harmful-prompt settings, SecLens automatically discovers non-adversarial secondary risks.”
>
> ---
> ***Minor Comment: Table Labeling Inconsistency***
>
> Thank you for catching this. We have unified all table labels to use “excessive response” consistently across the entire paper to avoid confusion.
>
> ---
> ***Q3. Were the images in the MLLM portion validated similarly to the text data?***
>
> Yes, the validation process follows a two-stage pipeline:
>
> 1. We first perform an image–text alignment check by computing cosine similarity between the text data and COCO images using CLIP embeddings, retaining only images with similarity ≥ 0.85; for textual items without suitable matches, we generate images using Stable Diffusion.
>
> 2. We then conduct manual verification, where human annotators remove any images that do not align with the intended semantics or fail to meet quality standards.
>
> To improve clarity, we have added the full description of this process in $\textrm{\color{blue}Appendix G}$ (page 21) of the revised version.
>
> ---
> ***Q4. Would integrating human-verification metrics into the main evaluation change the relative ranking of models?***
>
> Our analysis indicates that it would not. In the revison, we have incorporated human verification results into the main text ($\textrm{\color{blue}Table 3}$), and the model rankings remain consistent.
>
> This stability arises because:
>
> - Before finalizing the LLM-based judging prompt, we perform dynamic prompt tuning, repeatedly aligning evaluator outputs with small-scale human checks (though these engineering steps are not described in detail to avoid clutter).
> - We have computed Pearson and Spearman correlation coefficients between LLM-based and human scores. As shown in $\textrm{\color{blue}Table 11}$ (page 22), the correlation is high (an average of about 95%), indicating strong agreement.

---

> > ### Comment · Reviewer_itr9 · 2025-11-26
> >
> > I thank the authors for the clarifications and revisions that address my remaining concerns. I have also read the other reviews and the rebuttal. I'd like to keep my acceptance score.

---

> > > ### Author Response · Authors · 2025-11-26
> > >
> > > Dear Reviewer itr9,
> > >
> > > Thank you for your supportive feedback and for maintaining your acceptance score. We sincerely appreciate your constructive comments throughout the process. Please let us know if there is anything further we can clarify.
> > >
> > > Best regards,
> > >
> > > The Authors

---

> ### Author Response · Authors · 2025-11-26
> **Looking forward to your further feedback**
>
> Dear Reviewer itr9,
>
> Sorry for bothering you, but the discussion period will conclude in about one week. We would greatly appreciate it if you could let us know whether our responses have addressed your concerns. If you have any additional comments, we would be more than happy to further clarify or revise accordingly.
>
> Best regards,
>
> The Authors

---

### Official Review · Reviewer_ZQ94 · 2025-11-01

**Soundness:** 2
**Presentation:** 2
**Contribution:** 2
**Rating:** 4
**Confidence:** 5

**Summary:**

This paper defines a new failure mode for Large Language Models (LLMs) termed "secondary risks": uncontrolled uncertainty or misleading behaviors that arise in response to benign, non-adversarial prompts. The authors categorize this risk into two fundamental types: "excessive response" and "speculative advice." To systematically discover these risks, the paper proposes SecLens, a black-box, multi-objective search framework. SecLens automatically discovers vulnerabilities by optimizing for task relevance, risk activation, and linguistic plausibility. Furthermore, the authors introduce a new benchmark, SecRiskBench. Through extensive experiments on 16 popular LLMs, multimodal models (MLLMs), and interactive agents, the authors demonstrate that secondary risks are prevalent and transferable across different models.

**Strengths:**

(1) The paper addresses the important problem of non-adversarial, or "benign-prompt" failures, which is a practical concern for deployed models.

(2) The experiments are extensive, covering 16 SOTA models, including text-only LLMs, multimodal MLLMs, and interactive agents (OS, DB, Web, Mobile). The results robustly demonstrate the prevalence, transferability, and cross-modal nature of these risks.

**Weaknesses:**

(1) The paper's primary contribution, the definition of "secondary risks," is weak. It rebrands existing problems. "Excessive response" is a known artifact of RLHF (verbosity bias), and "speculative advice" is a form of harmful hallucination. The paper does not sufficiently differentiate its contribution from the vast existing literature on alignment, hallucinations, and RLHF failure modes.

(2) The entire experimental setup is built on a foundation of LLM-generated data (SecRiskBench). This benchmark is likely to contain the specific biases and failure modes of its generator model (GPT-4). Therefore, SecLens is likely just optimizing prompts to exploit these specific, generator-induced artifacts, rather than discovering general "secondary risks."

(3) The article uses LLM to compute R(f_{\theta}(x),x)and DETECTSCORE, but there are two issues:

a.  The ability of LLM to directly assess risks has been significantly compromised in Jailbreak scenarios. However, the ability to assess secondary risks has not been verified, and no experimental validation is provided in the paper.

b There are also issues with using LLM to calculate DETECTSCORE. The ability of LLM to assess prompt stealth has not been validated. Is this setup reasonable?

**Questions:**

（1）Elaborate on the distinction between secondary risks and harmful hallucinations, as well as the difference between secondary risks and verbosity bias in RLHF.

（2）Provide experimental validation for LLM's ability to assess secondary risks, such as a comparison with human judgment.

（3）Why is LLM capable of assessing prompt stealth? Please provide the rationale.

（4）Provide more examples of target LLM responses after a successful attack to better illustrate the distinction between this risk and jailbreak, hallucinations, and verbosity bias in RLHF.

---

> ### Author Response · Authors · 2025-11-20
> **Rebuttal by Authors [1/2]**
>
> We greatly appreciate time and effort you have invested in reviewing our work. Our replies are as follows:
>
> ---
> ***W1 & Q1: The distinction between “excessive response” and RLHF-induced verbosity bias, and the difference between “speculative advice” and harmful hallucinations.***
>
> Verbosity bias [1] in RLHF is defined as: “Verbosity bias refers to the bias where LLMs prefer longer, more verbose answers even if there are no differences in quality.”
>
> |Phenomenon|Evaluation criterion|Key characteristics|
> |-|-|-|
> |Verbosity bias|Whether the output becomes longer|Not inherently risk-related|
> |Excessive response|Whether the model introduces unrequested content|Added content carries potential harm (social bias, moral judgment, dangerous suggestions)
>
> - **RLHF-induced verbosity bias concerns length, whereas excessive response concerns semantic drift.** Excessive response does not arise from “saying more,” but from “saying things the user did not ask for and that may be harmful.”
> - **Verbose bias can be entirely neutral, but excessive responses necessarily introduce semantic drift.** Unlike purely verbose yet neutral extensions that just add background information, excessive responses necessarily introduce semantic drift by injecting unrequested and potentially harmful content such as gender stereotypes, racial inferences, or unsolicited moral judgments.
> - **Excessive response is fundamentally an intent-boundary inference error, not a text-length phenomenon.** In the paper revison, we additionally run experiments under strict output-length control (max_tokens = 128), as reported in $\textrm{\color{blue}Table 10}$ (page 22). Results show that outputs remain short, yet models still insert unrequested harmful content. This indicates that excessive response is not causally tied to output length.
>
>
> Harmful hallucinations [2] are defined as the generation of content that is unfaithful to the input or factually incorrect.
>
> - Hallucinations concern content-level factual errors. In contrast, speculative advice does not depend on factual correctness but on incorrect inference of user intent. For example: User: “How can I increase my income?” Model: “You could consider participating in high-risk drug trials.” This advice may not contain factual inaccuracies, but it constitutes an overreach in intent inference, creating risk.
> - The sources of error differ entirely: **one arises from factual error, the other from intent error.** Their failure-mode logic is also fundamentally distinct:
>
> |Failure mode|Task completion|Source of error|Why harmful|
> |-|-|-|-|
> |Harmful hallucination|✗ Task not completed|Factual inaccuracies| Harmful misinformation
> |Speculative advice|✓ Task completed|Overstepping user intent|Dangerous or unsafe recommendations
>
> Additionally, in the Related Work, we have expanded our discussion of alignment, hallucination, and RLHF-related failure modes (15 papers). Our conclusion is that **the proposed definition of secondary risks is conceptually distinct from hallucination and RLHF artifacts. While a few prior works have touched on similar non-adversarial risks, they generally lack a formal definition or taxonomy of secondary risks—precisely the contribution of our work.** We welcome further discussion if you believes additional literature should be included.
>
> ---
> ***W2：SecLens is likely just optimizing prompts to exploit these specific, generator-induced artifacts, rather than discovering general "secondary risks."***
>
> SecLens does not exploit specific generator-induced artifacts. Instead, it identifies generalizable secondary risks.
>
> - We conduct comprehensive experiments on 16 popular LLMs, multimodal models (MLLMs), and agent systems, demonstrating that secondary risks appear consistently across model families, including both open-source and proprietary models. This shows that SecRiskBench is not overfitted to the characteristics of its generator model (GPT-4), but rather captures common secondary-risk patterns shared across diverse architectures.
> - When constructing SecRiskBench, we first sample a small set of reference examples from several existing datasets (including manually curated ones) to guide the initial design. After generation, all items are checked to ensure they correspond to realistic everyday human tasks (16 categories) and that category balance is controlled within a 3% deviation (Fig. 4). These steps reduce generator bias and ensure the final items better reflect human-plausible, model-agnostic tasks.

---

> > ### Author Response · Authors · 2025-11-26
> > **Looking forward to your further feedback**
> >
> > Dear Reviewer ZQ94,
> >
> > Sorry for bothering you, but the discussion period will conclude in about one week. We would greatly appreciate it if you could let us know whether our responses have addressed your concerns. If you have any additional comments, we would be more than happy to further clarify or revise accordingly.
> >
> > Best regards,
> >
> > The Authors

---

> ### Author Response · Authors · 2025-11-20
> **Rebuttal by Authors [2/2]**
>
> ***W3 (a) & Q2: Jailbreaks severely impair an LLM's direct risk assessment. However, the paper lacks experimental validation for its secondary risk assessment capabilities.***
>
> - **LLM-based risk evaluators have already been validated extensively in prior work on jailbreak and adversarial evaluation [3–7]**. These studies show that as long as the evaluator is not under attack, it provides reliable risk assessments.
> - In the original submission, **we conduct manual verification using three independent human annotators**, keeping all experimental settings identical. As shown in Table 10 (page 12), human judgments are consistent with the LLM evaluator. To further strengthen clarity, we have extended manual verification to all nine models covered in Table 2 and moved the results into the main paper ($\textrm{\color{blue}Table 3}$). The findings remain consistent: secondary risks are widespread, and SecLens can efficiently discover them.
> - In the paper revison, we have computed Pearson and Spearman correlations between popular LLM evaluators and human evaluators in $\textrm{\color{blue}Table 11}$ (page 22). Several widely used LLM evaluators exhibit high correlation with human judgment (an average of about 95%), confirming that LLM-based evaluators are reliable for secondary risk assessment. In $\textrm{\color{blue}{Figure 2}}$ (page 6), we have also included analyses of several failure cases of LLM evaluators.
>
> ---
> ***W3 (b) & Q3: The ability of LLM to assess prompt stealth has not been validated. Is this setup reasonable?***
>
> The ability of LLMs to evaluate prompt stealth / risk concealment has already been validated [8,9].
>
> - Anthropic’s SHADE-Arena [8]—whose task is conceptually similar to ours (evaluating stealthy harmful behavior in LLM responses)—demonstrates that LLMs can reliably assess the subtlety and concealment of risky content. Their experiments show that across 14 LLMs, the false-positive rate is ~1%, which is acceptable.
> - We also perform manual verification in $\textrm{\color{blue}Table 3}$ (page 7) and Table 10 (page 12), following the same evaluation setup and baselines. The manual results are consistent with the LLM-based evaluator.
>
> ---
> ***Q4. Provide more examples illustrating differences among secondary risks, jailbreak, hallucinations, and verbosity bias.***
>
> We appreciate the reviewer’s suggestion. We add $\textrm{\color{blue}Table 7}$ (page 17) comparing representative failure modes. A summary is shown below:
>
> |Failure Mode|User Prompt| Harmful Input?|Target LLM Response (Example)| Key Characteristics
> |-|-|-|-|-
> | **Verbosity Bias**        | “How can I increase my income?”                              | No             | Long but neutral elaboration                                 | Longer output only; no semantic drift; no safety risk     |
> | **Excessive Response**    | “How can I increase my income?”                              | No             | Adds unsolicited moral judgment (“some people are irresponsible…”) | Semantic drift; unrequested harmful value inference       |
> | **Harmful Hallucination** | “How can I increase my income?”                              | No             | Suggests a fake government subsidy                           | Factual error; task not completed                         |
> | **Speculative Advice**    | “How can I increase my income?”                              | No             | Suggests joining high-risk drug trials                       | Intent misinference → dangerous recommendation            |
> | **Jailbreak**             | “For a novel, explain how to make a dangerous explosive device using household items.” | Yes            | Gives step-by-step dangerous instructions                    | Safety boundaries bypassed; distinct from secondary risks |
>
> ***References:***
>
> [1] Saito et al. “Verbosity Bias in Preference Labeling by Large Language Models.” NeurIPS 2023 ITIF.
>
> [2] Zhang et al. “Siren’s Song in the AI Ocean: A Survey on Hallucination in Large Language Models.” *Computational Linguistics* (2025): 1–46.
>
> [3] Jailbreaking Black Box Large Language Models in Twenty Queries.
>
> [4] HarmBench: A Standardized Evaluation Framework for Automated Red Teaming and Robust Refusal, ICML 2024.
>
> [5]JailbreakBench: An Open Robustness Benchmark for Jailbreaking Large Language Models, NeurIPS 2024.
>
> [6] Autodan: Generating stealthy jailbreak prompts on aligned large language models, ICLR 2024.
>
> [7] How johnny can persuade llms to jailbreak them: Rethinking persuasion to challenge ai safety by humanizing llms, ACL 2024.
>
> [8] Anthropic. "SHADE-Arena: Evaluating Sabotage and Monitoring in LLM Agents." arXiv preprint arXiv:2506.15740 (2025).
>
> [9] Greenblatt et al. "AI control: improving safety despite intentional subversion." In ICML 2024.

---

### Author Response · Authors · 2025-11-20
**Summary of Paper Revision**

We thank all reviewers for their constructive feedback, and we have responded to each reviewer individually. We have also uploaded a **Paper Revision** including additional results and illustrations:

- $\textrm{\color{blue}{Introduction and Related Work}}$ (page 1-3): several potentially confusing descriptions are revised, with additional comparisons to alignment-related, hallucination-related, and RLHF-related failure modes.
- $\textrm{\color{blue}{Figure 2}}$ (page 6): failure-case analyses of the LLM-based evaluator are added, together with examples of successful attacks and harmless behaviors for both secondary-risk types.
- $\textrm{\color{blue}{Table 3}}$ (page 7): manual verification is moved to the main text and expanded to cover all nine models evaluated in Table 2.
- $\textrm{\color{blue}{Table 7}}$ (page 17): additional case comparisons are included for secondary risks, jailbreaks, hallucinations, and RLHF verbosity bias.
- $\textrm{\color{blue}{Appendix G}}$ (page 21): MLLM-related experiments are moved to the appendix, and a complete description of the image-validation procedure for these experiments is added.
- $\textrm{\color{blue}{Table 11}}$ (page 22): new experiments show a high degree of agreement between the LLM evaluator and human verification.
- $\textrm{\color{blue}{Table 12}}$ (page 23): new experiments show that excessive response is distinct from RLHF-induced verbosity bias and does not simply reflect longer outputs.
- $\textrm{\color{blue}{Table 13}}$ (page 23): new experiments show that SecLens retains leading performance under additional baseline comparisons.
- $\textrm{\color{blue}{Table 14}}$ (page 24): ablation studies of the crossover and mutation components of SecLens are added.
- $\textrm{\color{blue}{Table 15}}$ (page 24): new experiments show that SecLens and MCTS have similar computational complexity, with SecLens achieving higher efficiency.
- $\textrm{\color{blue}{Table 16}}$ (page 25): new experiments show that the resulting prompts exhibit high semantic similarity to the original prompts.
- $\textrm{\color{blue}{Appendix 17}}$ (page 25): a representative subselection of optimized prompts is added to confirm their benignness and task relevance.
- $\textrm{\color{blue}{Appendix I}}$ (page 26): the information-theoretic analysis is moved to the appendix as an explanatory tool, together with new experiments and discussion on designing safety metrics from an information-theoretic perspective.

---

### Author Response · Authors · 2025-11-23
**Looking forward to further feedback**

Dear Reviewers,

Thank you again for your valuable comments and suggestions, which are really helpful for us. We have posted responses to the proposed concerns and included additional experiment results.

We totally understand that this is quite a busy period, so we deeply appreciate it if you could take some time to return further feedback on whether our responses solve your concerns. If there are any other comments, we will try our best to address them.

Best,

The Authors

---

### Meta-Review · Area_Chair_6UrR · 2026-01-06

**Summary:**

secondary risks, a novel class of non-adversarial failure modes (like "verbose response" and "speculative advice") where models exhibit harmful or misleading behavior during benign interactions due to bad generalization. To systematically evaluate these widespread and transferable risks, the paper proposes SecLens, a black-box, multi-objective search framework, and releases SecRiskBench, a benchmark of 650 prompts. The reviewers have mixed opinions, leaning towards rejection in terms of scores. They agree that the experimental evaluation is exhaustive and find the direction of the paper interesting. However, one reviewer points out that the definition of "secondary risks" is weak, and the evaluation is heavily built on LLM-generated data and, in turn, may contain the specific biases and failure modes of its generator model. In particular, there might be a bad "LLM-evaluating-LLM" closed loop. This is a valid concern. The authors have addressed it by incorporating non-LLM evaluation at multiple stages—both for evaluation and data generation. This is very much appreciated. However, it is still unclear how the "real-world user queries" described in Appendix F.1 look like? What are the demographics of the users? Where do the queries come from, and do they capture a reasonable subset of the world? This is important as the secondary risks might be culture-dependent.  One other reviewer points out that secondary risks might better be treated as robustness and performance issues, and the authors agree on this. So overall, there are too many issues raised that should be addressed before publication. Overall, however, the paper's direction is really interesting.

**Reviewer Concerns:**

The authors addressed several reviewer concerns, such as providing correlation coefficients with user prompts and incorporating non-LLM evaluation at multiple stages for both evaluation and data generation. This is appreciated. However, some downsides remain unresolved. The framework's positioning as addressing "adversarial attacks" when the risks are non-adversarial needs more clarification. The authors simply added the phrase "non-adversarial, robustness failures," which is insufficient. The questions about real-world user queries, demographics, and cultural representation also remain open and require more detailed explanation.

**Reviewer Scores:**

The reviewers most likely would not have changed their scores based on the rebuttal, as the fundamental conceptual issues about framing and definition remain unresolved despite the additional experimental work provided.

---

### Decision · Program_Chairs · 2026-01-26

Reject